# *Magnetospirillum magneticum* as a Living Iron Chelator Induces TfR1 Upregulation and Decreases Cell Viability in Cancer Cells

**DOI:** 10.3390/ijms22020498

**Published:** 2021-01-06

**Authors:** Stefano Menghini, Ping Shu Ho, Tinotenda Gwisai, Simone Schuerle

**Affiliations:** Department of Health Sciences and Technology, Institute for Translational Medicine, ETH Zurich, CH-8092 Zurich, Switzerland; stefano.menghini@hest.ethz.ch (S.M.); psho.cliff@gmail.com (P.S.H.); tinotenda.gwisai@hest.ethz.ch (T.G.)

**Keywords:** magnetotactic bacteria, iron chelator, cancer therapy, transferrin receptor 1, siderophores

## Abstract

Interest has grown in harnessing biological agents for cancer treatment as dynamic vectors with enhanced tumor targeting. While bacterial traits such as proliferation in tumors, modulation of an immune response, and local secretion of toxins have been well studied, less is known about bacteria as competitors for nutrients. Here, we investigated the use of a bacterial strain as a living iron chelator, competing for this nutrient vital to tumor growth and progression. We established an in vitro co-culture system consisting of the magnetotactic strain *Magnetospirillum magneticum* AMB-1 incubated under hypoxic conditions with human melanoma cells. Siderophore production by 10^8^ AMB-1/mL in human transferrin (Tf)-supplemented media was quantified and found to be equivalent to a concentration of 3.78 µM ± 0.117 µM deferoxamine (DFO), a potent drug used in iron chelation therapy. Our experiments revealed an increased expression of transferrin receptor 1 (TfR1) and a significant decrease of cancer cell viability, indicating the bacteria’s ability to alter iron homeostasis in human melanoma cells. Our results show the potential of a bacterial strain acting as a self-replicating iron-chelating agent, which could serve as an additional mechanism reinforcing current bacterial cancer therapies.

## 1. Introduction

Due to limited selectivity in systemically delivered cancer therapeutics, interest has grown in harnessing bacteria as living, tumor-targeting anticancer agents. The therapeutic potential of facultative anaerobic bacteria has been supported by studies demonstrating the delivery of non-pathogenic strains of *Escherichia coli* to solid flank tumors with associated tumor regression [1]. Additionally, safe administration of *Salmonella typhimurium* (VPN20009) has been shown for animal models and patients with metastatic melanoma [2,3]. Bacteria can act therapeutically by secreting innate or engineered toxins in situ (e.g., hemolysin E), transporting attached nanodrug formulations, or stimulating an immune response [4,5,6,7]. Colonizing bacteria can also engage in nutrient competition within the tumor microenvironment [8,9,10]. While the starvation of glucose as a crucial energy source to all cells has been studied extensively [11,12,13], other nutrients that are in specifically high demand by cancer cells might serve as more specific, vulnerable targets for deprivation.

Iron metabolism, for example, is significantly altered in mammalian tumor cells and recognized as a metabolic hallmark of cancer [14,15]. The main iron uptake mechanism adopted by most cells utilizes the internalization of transferrin receptor 1 (TfR1) upon binding of Fe (III)-bound transferrin (Tf). TfR1 expression positively correlates with cellular iron starvation and is upregulated in cancer cells, since malignant cells generally require a nutrient surplus [15,16,17]. Accordingly, several types of iron-scavenging molecules have been utilized to compete with malignant cells for available iron sources and have demonstrated significant antineoplastic activity both in vitro and in vivo [18,19,20]. Promising bacteria-derived iron-chelating siderophores, such as deferoxamine (DFO), as well as synthetic iron chelators have been developed for therapeutic purposes [21]. However, non-negligible side effects, including systemic toxicity and low efficacy, have hampered their translation into clinical trials as therapeutic agents for cancer treatment [22,23,24].

For this study, we investigated the potential of a specific bacterial strain with high demand for iron to serve as a local, self-replicating iron chelator that could thereby reduce systemic effects. Magnetotactic bacteria (MTB), like other bacteria, possess the ability to secrete high-affinity iron-scavenging siderophores. In particular, AMB-1 secrete both hydroxamate and catechol (3,4-dihydroxybenzoic acid) types of siderophores [25,26]. Unlike other bacteria, their demand for iron is particularly high, since this mineral is crucial for both their survival and the synthesis of unique intracellular organelles called magnetosomes [27,28]. These biomineralized magnetic nanocrystals are arranged in chains enclosed in a lipid bilayer and enable the bacteria to align with magnetic fields [29,30,31]. Furthermore, MTB are aerotactic, possessing an oxygen-sensing system that regulates motility in an oxygen gradient [32]. These features have previously been leveraged to magnetically guide MTB to the hypoxic core of solid tumors, yielding significantly higher tumor accumulation and penetration compared to their administration in the absence of external magnetic fields [33]. Once on site, nutrients from the tumor microenvironment are sourced to maintain proliferation and growth, and we hypothesize that MTB could induce iron deprivation of cancer cells.

To study this, we employed *Magnetospirillum magneticum* strain AMB-1 and first quantified the production of siderophores, benchmarked with molar concentrations of DFO. We then investigated the influence of AMB-1 on cell surface TfR1 expression using human melanoma cells and demonstrated the ability of AMB-1 to affect iron homeostasis. Finally, we examined the effect of AMB-1 on cancer cell growth in vitro by analyzing cell viability. The iron-scavenging capabilities of bacterial strains with naturally high or enhanced siderophore production may act as an additional mechanism for bacterial cancer therapy, complementing or augmenting established bacterial anticancer mechanisms.

## 2. Results

### 2.1. AMB-1 Proliferate and Produce Siderophores under Mammalian Cell Culture Conditions

First, we sought to examine the ability of AMB-1 to proliferate under mammalian cell culture conditions at 37 °C. Optical density (OD_600_) was measured on Day 0 (OD_600_ = 0.1417) and on Day 2 (OD_600_ = 1.808) to determine the concentration of AMB-1 in solution (Appendix A, Appendix A). To further ascertain the increased number of bacteria, the amount of MTB in Dulbecco’s Modified Eagle’s Medium (DMEM) was then quantified using a particle counting system. The recorded numbers of AMB-1 on Day 0 and Day 2 were 2.26 × 10^8^ and 3.36 × 10^9^, respectively (Appendix A). Lastly, bacterial viability following culture in DMEM was analyzed. A considerable increase in the number of viable bacterial cells could be observed after 48 h of incubation, confirming substantial growth of MTB at 37 °C (Appendix A).

Having established the ability of AMB-1 to proliferate in DMEM, we next determined to what extent AMB-1 produce siderophores in DMEM. Using the Chrome Azurol S (CAS) assay (Appendix A), 10^8^ AMB-1 bacteria were found to produce 0.10 ± 0.005 siderophore units in DMEM supplemented with 25 µM holo-transferrin (holo-Tf), while siderophore production in transferrin-free DMEM was negligible (Figure 1A). AMB-1 siderophore production was compared to the widely used iron chelator deferoxamine. It was found that the siderophores produced by 10^8^ AMB-1 in Tf-supplemented media were equivalent to 3.78 µM ± 0.117 µM deferoxamine (Figure 1B). These experiments demonstrated that AMB-1 could survive and proliferate at 37 °C and that the bacteria could produce a quantifiable amount of siderophores when holo-Tf was supplemented to the mammalian cell culture media.

### 2.2. AMB-1 Upregulates TfR1 Expression in Human Melanoma Cells

To assess whether AMB-1 can affect the iron uptake machinery of an aggressive cancer type, we co-cultured the bacteria with MDA-MB-435S, a malignant cell line established from M14 melanoma. We then monitored the bacteria’s effect on TfR1 expression using immunofluorescence. To mimic the tumor microenvironment, all experiments were performed under hypoxic conditions (Appendix A). The surface expression of TfR1 increased 2.7-fold on cancer cells co-cultured with live bacteria at AMB-1:MDA-MB-435S ratios as low as 10:1 (10^6^ AMB-1). The TfR1 upregulation was shown to increase with increasing bacteria ratios (Figure 2A,B). Deferoxamine was used here to create iron-deficient cell culture conditions as a positive control. MDA-MB-435S cells showed a significant and increasing upregulation of TfR1 surface expression up to 5.6-fold. To ensure that the upregulation of TfR1 expression was on the cell surface and not cytoplasmic, cell membrane integrity in the cultures was monitored. Less than 5% of cells were stained by the cell-impermeant DNA stain propidium iodide (PI), indicating cell membrane preservation over time (Figure 2C).

To gain insights on the TfR1 expression kinetics of the cell population, AMB-1-induced increase of cell surface TfR1 expression was analyzed over time. The effect at an AMB-1: MDA-MB-435S ratio of 1000:1 was already apparent after 6 h of co-culture (Figure 2D). The fluorescence intensity after 24 h of co-culture was 1.8 times higher than the initial value, while the change reached 95% of the final value after 12 h (Figure 2E). Untreated cancer cells did not display any increase in fluorescence (Figure 2F).

Upregulation of TfR1 could also not be detected for non-magnetotactic bacteria with lower demand for iron, such as *E. coli* [34,35,36]. Although *E. coli* Nissle 1917 have been shown to produce different types of siderophores under different environmental conditions [37], previous studies report that magnetotactic bacteria have a need for iron that can be up to 100 times higher compared to *Escherichia coli* cells [25,38]. When *E. coli* Nissle 1917 were incubated with melanoma cells at a ratio of 1000:1, our highest bacteria-to-cell ratio tested for AMB-1, no detectable increase in TfR1 expression could be noted on the cell surface of MDA-MB-435S cells (Appendix A). Altogether, these findings show that the bacterial strain AMB-1 possesses a unique ability to induce TfR1 upregulation in the tested human melanoma cancer cell line, thereby suggesting a direct link between AMB-1 induced disruption of iron uptake and TfR1 expression.

### 2.3. Reduced Viability of Cancer Cell Lines upon Incubation with AMB-1

Upon co-culturing melanoma cells with AMB-1 bacteria, we assessed cellular viability using an MTT assay. To establish that the assay would not unspecifically include the bacteria’s viability, an assay to ascertain the required number of washing steps to remove the bacteria was performed (Appendix A). A significant decrease in cell viability could be observed when cells were exposed to live bacteria at AMB-1:MDA-MB-435S ratios as low as 100:1 (10^7^ AMB-1). Incubation of MDA-MB-435S cells with bacteria (ratio 1000:1) resulted in an overall decrease of the mean cell viability of 62% (±21.93%) (Figure 3). To ascertain that this effect was not restricted to one cell line, the experiment was repeated on an additional type of invasive cancer cells. For this purpose, we used MDA-MB-231, a human breast adenocarcinoma cell line derived from a metastatic site. A significant decrease of 65% (±23.37%) was detected in MDA-MB-231 cell viability once the cells were incubated with bacteria at a ratio of 1000:1. Supported by these observations, we showed that magnetotactic bacteria AMB-1 impact cancer cell viability, suggesting that they affect cancer cell growth in vitro.

## 3. Discussion

AMB-1 are a strain of magnetotactic bacteria known to preferably grow at temperatures around 25–30 °C [39,40]. Alterations in proliferation rate are expected whenever deviations from their optimal growth conditions occur. Benoit et al. (2009) have previously reported the ability of AMB-1 to still reproduce and form magnetite when cultured at 37 °C (in vitro and in vivo) [41], a finding we here independently corroborated by investigating AMB-1 proliferation and viability over 48 h at 37 °C (Appendix A). We determined the concentration of bacteria in suspension by measuring optical density and then confirmed these data by quantifying the bacterial cells at different intervals (Appendix A). We could further validate the increased number of bacterial cells and ascertain the presence of live AMB-1 after two days of incubation using a Live/Dead stain (Appendix A). We next investigated the bacteria’s ability to produce iron-chelating molecules at 37 °C. We quantified the number of siderophores produced by the *Magnetospirillum magneticum* strain AMB-1 in mammalian cell culture medium and benchmarked the results with deferoxamine, a commonly used iron chelator. Taken together, our findings show that AMB-1 still proliferate and produce siderophores when cultured in mammalian cell culture medium at 37 °C.

We then showed that AMB-1 inoculation with human melanoma cell cultures affects iron homeostasis of the cancer cells. Iron homeostasis is essential for normal cell growth and development, and iron starvation is mainly characterized by alterations in the iron import machinery, specifically by an upregulation of the transferrin receptor 1 on the cell surface. Increased TfR1 expression found on MDA-MB-435S melanoma cancer cells correlates with increasing bacteria ratios, suggesting that AMB-1 limit iron availability to the mammalian cells (Figure 2A,B). A significant increase of TfR1 expression could already be detected 6 h after inoculation (Figure 2D–F). The observed increase occurred in a nonlinear manner, which may be due to an energetic trade-off [42]. We hypothesize that at lower bacterial concentrations the cells would mainly spend metabolic energy on the upregulation of TfR1, whereas at increasing concentrations of AMB-1 the energy might be shifted towards responses to the accumulation of foreign microorganisms in the environment. Therefore, a nonlinear increase in TfR1 expression might not be unexpected when the ratio of bacteria to cancer cells is increased. Similarly, the cancer cells showed a significant upregulation of TfR1 surface expression after incubation with deferoxamine (10 µM and 25 µM), in line with previous reports on cellular iron deficiency [15,16,43]. These observations demonstrated that AMB-1 affects the iron import mechanisms of human melanoma cells, acting as an effective competitor for iron when in co-culture with MDA-MB-435S cells.

Furthermore, we assessed the impact of AMB-1 cells on cancer cell growth (Figure 3). Earlier studies indicated the benefits of adding iron chelators to cancer cells, showing a reduction of cell growth upon treatment [19,24,43]. Our results confirmed that an increasing number of AMB-1 added to co-culture correlated with a decrease in the percentage of viable cancer cells. This effect could be detected on cancer cells from two different lineages. At the highest investigated bacteria-to-cell ratio (1000:1), the melanoma cells MDA-MB-435S displayed an overall viability of 38% and the viability of breast cancer cells MDA-MB-231 was 32%. These findings suggest that the high requirement for iron exhibited by MTB causes them to actively compete for Fe (III) with cancer cells, leading to a nutrient shortage associated with reduced viability.

Our data support the idea that AMB-1 have the ability to act as living iron chelators by secreting a quantifiable amount of siderophores. We showed that 10^8^ AMB-1/mL can produce high-affinity iron-scavenging molecules equivalent to 3.78 µM deferoxamine over 24 h (Figure 1B). Previous works demonstrated that the treatment of different cell lines with 10 µM–30 µM deferoxamine significantly reduced cell viability in vitro [19,43]. Moreover, a significant diminution of cell viability was even detected at the lower deferoxamine concentration of 2.5 µM when combined with the chemotherapeutic drug cisplatin [19]. Nonetheless, the implementation of molecular iron-scavenging molecules in translational medicine is hampered by elevated systemic toxicity, as well as limited tumor selectivity. These challenges might be overcome by implementing bacteria as direct competitors for nutrients at the tumor site. Several approaches have been investigated in the past to deliver magnetotactic bacteria to solid tumors [44]. For example, intravenously introduced AMB-1 have been shown to colonize 293T tumor xenografts 3–6 days after injection [41] without external magnetic guidance and act as a local T2-weighted contrast imaging agent. Another study described the increased accumulation of magnetotactic bacteria strain MC-1 in HCT116 colorectal xenografts upon peritumoral injection, when exposed to external guiding magnetic fields [33]. Moreover, the ability of swarms of AMB-1 to enrich and penetrate dense model tissue matrices when powered by external rotating magnetic fields was recently demonstrated [45]. The unique trait of magnetotactic bacteria to respond to magnetic fields, both for imaging and control, spurred interest in their investigation as a powerful addition in current attempts of bacterial cancer therapy. Here, in particular, solid tumors possess a characteristic that renders them very appealing since they are characterized by a hypoxic core that possesses the ability to provide a niche for anaerobic bacteria, such as AMB-1 [46]. Tumor-targeting bacteria offer unique therapeutic options to suppress cancer, such as local production and delivery of anticancer agents through genetic manipulation and initiating antitumor immune responses [7]. While so far mostly commensal non-magnetic bacterial strains have been investigated, information on the immune response to magnetotactic bacteria remains limited. It has been demonstrated, however, that intratumoral injection of chains of their extracted magnetosomes coated with endotoxins elicits recruitment of immune cells to the tumor site and triggers cancer regression [47]. Furthermore, the intrinsic magneto-aerotactic capability of MTB allows them to regulate their motility towards environments with low oxygen concentration and react to externally applied magnetic fields [31,32]. Aerotaxis and anaerobic traits have also been leveraged in other strains, such as *Salmonella*, enabling them to act as bacterial anticancer agents that target necrotic tumor microenvironments with poor oxygen supply [48,49,50]. Overall, the intrinsic abilities of AMB-1 to self-replicate, respond to magnetic fields, and secrete sustained doses of siderophores warrant further study in the context of cancer therapy. By combining the benefits of bacterial cancer therapy with iron chelation and other traits of AMB-1, we envision that magnetotactic bacteria could become a valid therapeutic agent to implement against cancer.

Our work motivates the use of living AMB-1 as self-replicating iron-scavenging organisms actively competing for this vital nutrient, with the possibility of compromising the survival of cancer cells. Further application could include the use of tumor-targeting organisms both as a monotherapy and as a combination therapy with established antineoplastic drugs to obtain optimal clinical outcomes. Although MTB are considered as non-pathogenic, unlike most of the bacteria currently tested for cancer therapy, they have been rarely studied in vivo until now and more studies are required to advance the knowledge about their adaptability to different environments and conditions in vivo [44]. However, the unique characteristics of magnetotactic bacteria could also be exploited to engineer iron-scavenging strains of surrogate commensal and attenuated bacteria that have already been established as anticancer agents [3,7]. This work lays the foundation for future investigations which combine iron chelation with bacterial cancer therapy to enhance existing therapeutic strategies and open new frontiers for combating cancer.

## 4. Materials and Methods

### 4.1. Bacterial Strain and Culture Condition

*Magnetospirillum magneticum* AMB-1, a strain of magnetotactic bacteria, was purchased from ATCC (ATCC, Manassas, VA, USA). AMB-1 bacteria were grown anaerobically at 30 °C, passaged weekly and cultured in liquid growth medium (ATCC medium: 1653 Revised Magnetic Spirillum Growth Medium). *Magnetospirillum magneticum* Growth Media (MSGM) contained the following per liter: 5.0 mL Wolfe’s mineral solution (ATCC, Manassas, VA, USA), 0.45 mL resazurin, 0.68 g of monopotassium phosphate, 0.12 g of sodium nitrate, 0.035 g of ascorbic acid, 0.37 g of tartaric acid, 0.37 g of succinic acid, and 0.05 sodium acetate. The pH of the media was adjusted to 6.75 with sodium hydroxide (NaOH) and then sterilized by autoclaving at 121 °C. Then, 10 mM ferric quinate (200×) Wolfe’s Vitamin Solution (100×) (ATCC, Manassas, VA, USA) were added to the culture media shortly before use. The concentration of AMB-1 in solution was determined by optical density measurement (Spark, Tecan, Männedorf, Switzerland) and the approximate number of bacteria was extrapolated from a standard curve.

Start of *E. coli* cultures was achieved by picking single colonies from LB agar plates and subsequent inoculation in Lysogeny broth (LB). Preculture of bacteria was performed the day before the experiment in liquid LB media overnight at 37 °C on a shaking device. On the day of the experiment, an approximate number of *E. coli* in solution was then determined by optical density measurement (Spark, Tecan, Männedorf, Switzerland),

### 4.2. CAS Assay to Assess Siderophore Quantification

*Magnetospirillum magneticum* AMB-1 were cultured in 1.7 mL phenol red-free DMEM (11054020, Invitrogen, Carlsbad, CA, USA) supplemented with GlutaMAX (35050061, Invitrogen, Carlsbad, CA, USA) in a sealed 1.5 mL Eppendorf tube at 37 °C for 48 h. Fetal bovine serum (FBS, Biowest, Nuaille, France) was excluded from the media and replaced with a known concentration of iron source, i.e., 25 μM holo-transferrin (T0665, Sigma-Aldrich, St. Louis, MO, USA). Quantification of siderophores produced by AMB-1 was performed using a Chrome Azurol S (CAS) assay (199532, Sigma-Aldrich, St. Louis, MO, USA) [51]. Then, 100 μL of each sample’s supernatant were collected and mixed with 100 μL CAS assay solution on a transparent 96-well plate. The assay was then incubated in the dark at room temperature for 1 h before the absorbance was measured at 630 nm on a multimode microplate reader (Spark, Tecan, Männedorf, Switzerland). The measurement was expressed in siderophore production unit (s.p.u.), which was calculated as follows:Siderophore production unit (s.p.u.) = (OD_630,ref_ − OD_630_)/OD_630,ref_(1)

DMEM supplemented with different concentrations of deferoxamine mesylate salt (DFO, D9533, Sigma-Aldrich, St. Louis, MO, USA) was prepared by serial dilution and used to generate a calibration curve (Appendix A).

### 4.3. Mammalian Cell Culture

Human melanoma MDA-MB-435S cells (ATCC, Manassas, VA, USA) and human breast cancer MDA-MB-231 cells (ATCC, Manassas, VA, USA) were cultured in high glucose Dulbecco’s Modified Eagle’s Medium (DMEM, Invitrogen, Carlsbad, CA, USA) supplemented with 10% fetal bovine serum (FBS, Biowest, Nuaille, France) and 1% penicillin/streptomycin (CellGro, Corning, NY, USA). All cells were incubated at 37 °C in a humidified atmosphere with 5% CO_2_.

### 4.4. Co-Culture of Mammalian Cancer Cells with Magnetotactic Bacteria

Human melanoma MDA-MB-435S cells (1 × 10^5^ cells) were cultured on 12-well plates and incubated in a 5% CO_2_ incubator at 37 °C for 24 h. For microscopic analysis at high magnification (>40×), a circular cover slip was placed in each well prior to cell seeding. Following incubation, *Magnetospirillum magneticum* AMB-1 (1 × 10^6^ to 1 × 10^8^ cells) were introduced into the wells. The well plate was stored in a sealable bag and the bag was flushed with nitrogen for 15 min in order to produce hypoxic conditions. The setup with the 12-well plate was then incubated at 37 °C for 48 h. To serve as negative and positive controls, 0, 10 µM, and 25 µM of the iron-chelating agent deferoxamine mesylate (D9533, Sigma-Aldrich, St. Louis, MO, USA) were added to the MDA-MB-435S cell culture in place of AMB-1 bacteria.

### 4.5. Immunofluorescence Labelling of MDA-MB-435S Cells

After the co-culture, cells were washed with ice-cold 1X Dulbecco’s Phosphate-Buffered Saline solution (DPBS, Gibco, Carlsbad, CA, USA) and then blocked with a 1% Bovine Serum Albumin (BSA, Sigma-Aldrich, St. Louis, MO, USA) solution diluted in 1X DPBS. The cells were then incubated with 10 µg/mL primary anti-TfR1 antibody (ab84036, Abcam, Cambridge, UK) on ice in the dark for 1 h. Subsequently, the cells were washed with ice-cold DPBS and incubated with 20 µg/mL secondary goat anti-rabbit antibody (ab150077, Abcam, Cambridge, UK) and 25 µg/mL Hoechst 33342 (H3570, Thermo Fisher Scientific, Waltham, MA, USA) on ice in the dark for another hour. Next, the cells were washed with ice-cold 1X PBS twice and fixed with a 2% paraformaldehyde (PFA) solution. Fixed cells were washed three times with 1X DPBS and the cover slips were mounted on glass slides and stored overnight in the dark at 4 °C. A Nikon Eclipse Ti2 microscope (Nikon Instruments, Tokyo, Japan) equipped with a Yokogawa CSU-W1 Confocal Scanner Unit (Yokogawa, Tokyo, Japan) and Hamamatsu C13440-20CU ORCA Flash 4.0 V3 Digital CMOS camera (Hamamatsu photonics, Hamamatsu, Japan) were used for visualization. Microscope operation and image acquisition was performed using Nikon NIS-Elements Advanced Research 5.02 (Build 1266) software (Nikon Instruments, Tokyo, Japan). ImageJ v2.0 (NIH, Bethesda, MD, USA) was used to process the obtained images.

### 4.6. Evaluation of Fluorescently Labelled MDA-MB-435S Cells by Flow Cytometry

Flow cytometry was used to measure the expression of fluorescently labelled TfR1 on the surface of MDA-MB-435S cells. Cells were harvested at different time points during co-culture (0 h, 6 h, 12 h, 24 h) and washed in cold 1X DPBS. Harvested cells were stained with primary anti-TfR1 antibody (ab84036, Abcam, Cambridge, UK) at a concentration of 10 µg/mL. After 1 h of incubation on ice, cells were washed twice with 1X DPBS and then stained with 20 µg/mL secondary goat anti-rabbit antibody (ab150077, Abcam, Cambridge, UK). Finally, cells were washed twice with 1X DPBS and analyzed by flow cytometry with a BD LSRFortessa device (BD Biosciences, San Jose, CA, USA) using a 488 nm excitation laser and 530/30 and 690/50 band pass emission filters for detection. FlowJo^TM^ (FlowJo LLC, Ashland, OR, USA) software was used to evaluate the data.

Flow cytometry was used to assess the cell membrane integrity of MDA-MB-435S cells. Cells were harvested at different time points during co-culture (0 h, 6 h, 24 h) and washed in cold 1X DPBS. Collected cells were stained with 1 µg/mL propidium iodide (V13242, Thermo Fisher Scientific, Waltham, MA, USA) and incubated for 30 min in a humidified atmosphere with 5% CO_2_ at 37 °C. Finally, cells were washed twice with 1X DPBS and analyzed by flow cytometry with a BD LSRFortessa device (BD Biosciences, San Jose, CA, USA) using a 488 nm excitation laser and 610/10 bandpass emission filters for detection. FlowJo^TM^ (Tree Star) software was used to evaluate data and graphs were plotted using Prism 8.0 (GraphPad, San Diego, CA, USA).

### 4.7. Investigation of Cell Viability Using an MTT Assay

A CyQUANT MTT Cell Viability Assay (V13154, Thermo Fisher Scientific, Waltham, MA, USA) was used to measure the viability of human melanoma cells MDA-MB-435S and human breast cancer cell lines MDA-MB-231. Cells were plated in 96-well culture plates (50,000 cells/well) and incubated in a 5% CO_2_ incubator at 37 °C for 24 h. Co-culture was then performed by adding AMB-1 bacteria at different ratios (10:1, 100:1, 1000:1) under hypoxic conditions, as described earlier. After 24 h of incubation, cells were washed 3× with cold DPBS to remove the bacteria. Next, 100 µL of DMEM and 10 µL of MTT stock solution (12 mM) were added to the wells and cells were then incubated at 37 °C for 4 h. The media was removed, and formazan crystals formed by the cells were dissolved in 50 µL DMSO. The absorbance was measured at 540 nm using a multimode microplate reader (Spark, Tecan, Männedorf, Switzerland). Background signal was subtracted from the final values and data were first normalized to an untreated control and then plotted as a percentage of the untreated cells.

An MTT viability assay was additionally used to investigate the number of washing steps required to remove AMB-1 from the wells. AMB-1 were added to the wells of a 96-well culture plate at a concentration corresponding to the number of bacteria used for the co-culture experiments. The well plate was kept under hypoxic conditions in an incubator at 37 °C. After 24 h, bacteria were either washed 0× or 3× with DPBS. Next, 100 µL of media and 10 µL of MTT stock solution (12 mM) were added to the wells and cells were then incubated at 37 °C for 4 h. The media was removed, and formazan crystals formed by the cells were dissolved in 50 µL DMSO. The absorbance was measured at 540 nm using a multimode microplate reader (Spark, Tecan, Männedorf, Switzerland). Background signal was subtracted from the final values.

### 4.8. Quantification and Staining of AMB-1 Bacteria

AMB-1 were grown anaerobically without agitation at 30 °C. Bacteria were split at a ratio of 1:10 and cultured for 1.5 days, to coincide with the proliferative phase (log phase). AMB-1 were then centrifuged at 9383 RCF for 10 min and the pellet was resuspended in cell culture media (DMEM supplemented with 10% fetal bovine serum). Additional tubes of bacteria were pelleted, resuspended in cell culture media, and incubated at 37 °C for 48 h. Excess volume was used to avoid trapping air in the tubes. Optical densities of the bacteria were measured in culture media. Additionally, 1 mL of media was measured separately and used for background subtraction. OD measurements were determined on Day 0 and Day 2 at a wavelength of 600 nm (OD_600_) using a multimode microplate reader (Spark, Tecan, Männedorf, Switzerland).

At both time points, bacteria were collected, pelleted, and stained using a BacLight viability Kit (L13152, LIVE/DEAD BacLight Kit, Thermo Fisher Scientific, Waltham, MA USA), according to the manufacturer’s protocol. Stained bacteria were added into Polydimethylsiloxane (PDMS) rings (Ø = 6 mm) and imaged using 100× magnification. Visualization and image acquisition were performed using confocal microscopy (Nikon Eclipse Ti2). ImageJ v2.0 (NIH) was used to process the obtained images.

Adopting the same experimental procedure, bacteria were collected and quantified on Day 0 and Day 2 using a particle counting system, Multisizer 4e Coulter Counter (Beckman Coulter, Brea, CA, USA).

### 4.9. Statistics and Data Analysis

All graphs and statistical analyses were generated using Prism 8.0 (GraphPad). Statistical significance and number of replicates of the experiments are described in each figure and figure legend. Error bars, where present, indicate the standard error of the mean (SD). *p*-values are categorized as * *p* < 0.05, ** *p* < 0.01, and *** *p* < 0.001.

## Figures and Tables

**Figure 1 ijms-22-00498-f001:**
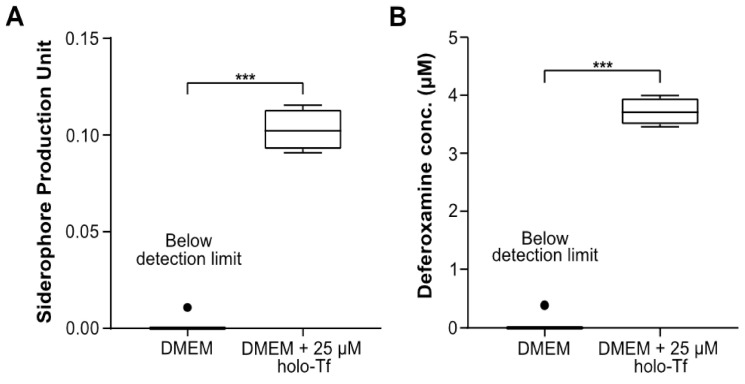
Quantification of siderophores produced by *Magnetospirillum magneticum* AMB-1 and analysis of their interaction with human transferrin (Tf). (**A**) Siderophores produced by AMB-1 were quantified by a Chrome Azurol S (CAS) assay in DMEM (condition 1) and DMEM supplemented with 25 μM holo-transferrin (holo-Tf) (condition 2) (*n* = 4 per condition, statistical significance was assessed with an unpaired two-tailed *t*-test, *** *p*-value < 0.001) (**B**) Siderophore production units plotted in terms of the inferred equivalent concentration of deferoxamine (DFO) (*n* = 4 per condition, statistical significance was assessed with an unpaired two-tailed *t*-test, *** *p*-value < 0.001).

**Figure 2 ijms-22-00498-f002:**
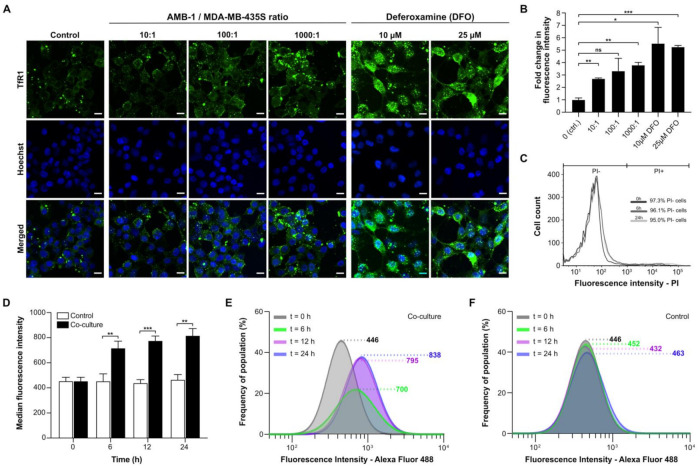
Analysis of transferrin receptor 1 (TfR1) upregulation and cell surface expression on MDA-MB-435S. (**A**) Representative immunofluorescence images of human melanoma cells co-cultured under hypoxic conditions for 48 h with different ratios of AMB-1 bacteria and different concentrations of deferoxamine as a positive control. Images show MDA-MB-435S cells marked by anti-TfR1 antibody (green) and Hoechst 33342 (blue), scale bar: 10 µM. (**B**) Graphical representation of the fluorescence intensities of the images shown in Figure 2A. Quantification of the fold changes in fluorescence were displayed relative to the control condition (*n* = 2 biological replicates per condition, statistical significance was assessed with an unpaired two-tailed *t*-test, * *p-*value < 0.05, ** *p-*value < 0.01, and *** *p-*value < 0.001). (**C**) Membrane integrity was measured as a graphical representation of propidium iodide (PI)-negative and PI-positive cell populations after 0, 6, and 24 h. (**D**) TfR1 median fluorescence intensity measured over 24 h, (*n* = 3 biological replicates per time point, statistical significance was assessed with an unpaired two-tailed *t*-test, ** *p-*value < 0.01, and *** *p-*value < 0.001). (**E**) Representative log-normal fitted fluorescence intensity histograms of cell surface TfR1 expression on MDA-MB-435S cells in co-culture model and (**F**) negative control, (*n* = 3 biological replicates per time point).

**Figure 3 ijms-22-00498-f003:**
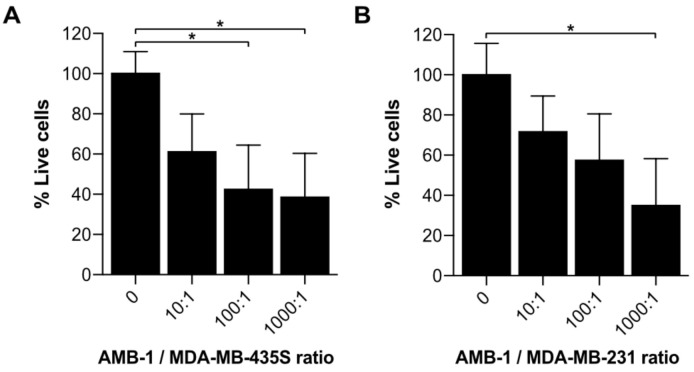
Investigation of cell growth upon incubation with AMB-1 bacteria. Cell viability of (**A**) MDA-MB-435S and (**B**) MDA-MB-231 was determined using an MTT assay and expressed as a percentage of the untreated cells. Viability (%) is expressed as mean ± SD of 3 individual biological replicates. Ordinary one-way ANOVA test was used to assess statistical significance (* *p*-value < 0.05).

## Data Availability

The data presented in this study is contained within the article and the supplementary material.

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
