# Peer review of "Magnetospirillum magneticum* as a Living Iron Chelator Induces TfR1 Upregulation and Decreases Cell Viability in Cancer Cells"

_ijms, 2021, doi:10.3390/ijms22020498_

Round 1

Reviewer 1 Report

Dear Authors,

the scientific work is clear and well presented. I appreciate your results clearly illustrated. To improve the discussion, I would reccomend a description of how bacteria could be employed in tumours, making some examples of what types of tumours could be addressed with these bacteria. It would also helpful an analysis of pros and cons of the Magnetospirillum magneticum employement as a cancer theraphy.

Best regards

Reviewer 2 Report

In this manuscript, Menghini et al. assessed the potential of the magnetotactic bacterium Magnetospirillum magneticum to act as a self-replicating iron-chelating agent in co-culture with cancer cells. Therefore, the authors determined the ability of M. magneticum to form siderophores. Additionally, co-cultivation experiments were conducted after which the cell viability and the abundance of the transferrin receptor 1 (TFR1) on the surface of a human cancer cell line were determined. It was found that M. magneticum produced up to 3.8 µM deferrioxamine equivalents in DMEM medium. Since cancer cells showed an upregulation of TFR1 and a reduced cell viability in the presence of M. magneticum the authors concluded that M. magneticum cells decrease the amount of available iron for cancer cells thereby inducing iron starvation and decreasing viability.

Although the concept of this manuscript is interesting, the experimental data is not sufficient to justify the author’s conclusions.

First, I have doubts about the fitness and viability of M. magneticum in the DMEM cell culture medium. This medium contains only minute amounts of carbon source that is metabolizable by M. magneticum (i.e. 1 mM pyruvate, all other components of DMEM including glucose are not used) (Matsunaga et al. 1991, https://www.thermofisher.com/de/de/home/technical-resources/media-formulation.52.html). Additionally, cultivations were performed at a temperature of 37°C, which is also beyond the optimal growth temperature for M. magneticum. To my experience, growth of magnetospirilla is extremely susceptible to pertubations and deviations from their optimal growth conditions. Thus, I cannot imagine that M. magneticum grows well in conditions used in this manuscript. It is also not clear if under these conditions magnetosome formation is still working in the extremely iron-poor DMEM. If not, the iron demand of M. magneticum would drop to levels of normal non-magnetic bacteria and cancer cells might not be inhibited by iron depletion. Thus, electron microscopic pictures before and after co-cultivation need to be presented.

In Fig. S 5. A the quantification and visualization of proliferation and viability of M. magneticum might argue for good growth of M. magneticum in DMEM over 48 hours at 37 °C. However, the quality of the provided microscopic pictures is too low to clearly discern spirally shaped cells of M. magneticum. Moreover, in these pictures I can only see huge structures of ~17 µm diameter which clearly exceed the size of normal M. magneticum cells (0.5 µm x 3 µm), single fluorescent dots or rod–like structures. Therefore, it is not clear for me that the presented pictures really show cells of M. magneticum. I highly recommend to use a 1000-fold magnification for the visualization of bacterial cells. The 25 µm (not µM as given in the manuscript) scale bar, however, suggests that the authors used a 400-fold magnification which they also used for the visualization of the much larger cancer cells. In summary, the different structures seen in the micrographs raise questions about the correctness of the presented cell numbers in Fig S5 A and, thus, viability of M. magneticum. Furthermore, due to the chosen scale of the Fig. S5 A Y-axis it is not immediately clear how many cells were used as inoculum (I also couldn’t find this number within the text).

Without providing  clear evidence that the bacterial cells are growing or at least viable in DMEM one cannot be sure that the increase of CAS-assay absorbance (Fig. 1) is caused by excretion of siderophores or release of cellular constituents by dying bacterial cells.

The authors also claimed that the excretion of siderophores by M. magneticum converts the iron-bound holo-transferrin of the DMEM medium to iron-free apo-transferrin based on the band width after conventional SDS-PAGE (Fig. S2). Recently, however, it was published that apo-and holo-transferrin cannot be discriminated using SDS-PAGE analyses (Ishikawa et al. Scientific Reports (2019) 9:10566) since iron is removed by denaturation of the holo-transferrin. Therefore, both forms have the same electrophoretic mobility and cannot be discriminated. The differences between holo- and apo-transferrin provided in Fig. S2, thus, appear to be related to different protein concentrations rather than the presence of bound iron molecules.

In Fig. 2 the authors show the increase of the TFR1 abundance in response to co-cultivation with M. magneticum. While the FACS-based data clearly shows a slight increase in fluorescently labeled TFR1. The corresponding fluorescent pictures of Fig 2A appear to be quite similar between the control and the 10:1 and 100:1 ratios. Here, the authors might analyze and compare average pixel intensities from fluorescent pictures to clarify this issue. Furthermore, I think a second control is required to show that the used antibodies do not (unspecifically) bind to bacterial cells as well. Such an artifact might also explain the increase in TFR1 immuno-fluorescence with increasing bacterial cell numbers. This control should also be done for every time point since the composition of the bacterial surface, and therefore antibody reactivity, might change over time.

In the manuscript, I missed an explanation for the non-linear increase of the TFR1 immuno fluorescence with the increasing bacterial cell number. I.e. while bacterial cell numbers increase 10 to 100-fold the increase in TFR1 immuno fluorescence is just from ~2.7-fold to 3.7-fold (estimated as exact numbers are not given in the text).

Furthermore, the authors tested an E. coli strain in a similar co-culture experiment and didn’t observe an increase in TFR1 immuno-fluorescence, even at a 1000:1 ratio. Strangely, the used E coli strain Nissle 1917 is able to produce a number of different siderophores over a wide range of environmental conditions (Valdebenito et al International Journal of Medical Microbiology 2006: 296(8) 513-520). Thus, one would expect similar results for siderophore-producing E. coli strains. Since this was obviously not the case, it remains to be investigated if the E. coli strain indeed formed siderophores upon DMEM cultivation using the CAS-Assay. Furthermore, as an additional control, bacterial strains definitely unable to produce siderophores should be tested as well (e.g. Magnetospirillum gryphiswaldense) to test for potential effects due to cell lysis, excretion of other non-siderophore substances or simple reduction of Fe3+ to Fe2+. Only then conclusions might be drawn from the co-cultivation experiments.

If the co-cultivation indeed affected the iron metabolism of the cancer cell line further changes in protein expression or a reduced cellular iron content should be observable. The authors, therefore, need to confirm their hypotheses with additional experiments that independently support the disrupted iron metabolism.

In summary, the manuscript in its current form has too many shortcomings, lacks proper controls and thus conclusions are not supported by the provided data. The manuscript therefore should not be accepted for publication in the International Journal of Molecular Sciences.

Round 2

Reviewer 2 Report

The revised version of the manuscript “Magnetospirillum magneticum to act as a self-replicating iron-chelating agent in co-culture with cancer cells“ by Menghini et al. significantly improved the quality of the study. However, some of the newly presented data need further explanation.

For example, the authors have now more convincingly shown that AMB-1 cells are viable in the tested cell culture medium. Nevertheless, it should be better described how the bacterial cells were precultivated and treated before inoculation into the cell culture medium. The current version still lacks a description of the conditions for precultures that were used as inoculum for the final experiment (e.g. temperature, agitation, O2 concentration, time, final OD/cell number) or if cells were washed in cell culture medium before inoculation. Next, I was surprised to see that AMB-1 can even grow to an OD of 1.8 in the tested cell culture medium whereas similar ODs and cell concentration where not or only hardly achievable with AMB-1 using dedicated/specialized fermentation systems (Olszewska-Widdrat et al. Reducing Conditions Favor Magnetosome Production in Magnetospirillum magneticum AMB-1 (2019); Yang et al. Effects of growth medium composition, iron sources and atmospheric oxygen concentrations on production of luciferase-bacterial magnetic particle complex by a recombinant Magnetospirillum magneticum AMB-1 (2001); Yang et al. Iron feeding optimization and plasmid stability in production of recombinant bacterial magnetic particles by Magnetospirillum magneticum AMB-1 in fed-batch culture (2001)). Maybe the authors could comment on their finding?

The newly provided micrographs of Fig. S2 are significantly better than those of the previous version. However, the micrograph from day two suffers from the high number of cells which doesn’t allow identification of single cells. I would therefore recommend to include an extra panel with a micrograph from a diluted sample of day 2.

Having shown that AMB-1 cells can grow in the tested cell culture medium and apparently produce siderophores under these conditions, the question arises if siderophores are still produced under co-culture conditions. This is especially interesting as the authors themselves mentioned that eukaryotic cell recognize and respond to the presence of bacterial cells (Gasque et al. 2015). As the presence of siderophores is crucial for the overall conclusions of this manuscript this question needs to be answered. Similarly, I still believe that siderophore production of the used E. coli strain needs to be assessed as well.

Further comments:

The experimental design for Fig. S5 needs some further explanation. E.g. were AMB-1 cells cultivated in DMEM for this assay or just briefly added and then washed? The readout for the MTT assay is given as RFU whereas the description for the MTT assay on page 10 of the main manuscript states that the absorbance at 540nm and not fluorescence was measured.

Results section 2.1. I would recommend to present viability data first.

On Page 4 the ratio of bacterial cells to cancer cells should be provided in a unified manner as 1000:1 or 1:1000 but not mixed (see x-axis of figure 2B). Furthermore, why didn’t the authors choes a deferrioxamine concentration that is closer to the amount produced by AMB-1?

On page 5 I also missed a brief description about the used cell lines (which type of cancer, why used here). This information can be partially found in other (later) parts of the manuscript but should be presented here when the cell lines are introduced for the first time.

The first sentence of the discussion (page 6) in its current form is not true as there are several studies published which show that MTB have multiple iron uptake systems that do not involve siderophores (e.g. Rong et al. 2008 and 2012).

Species names like E. coli should be written in italics throughout the whole manuscript (e.g. page 5).

The manuscript contains a few typos which should be removed for publication. Please check.
